# Nintedanib-αVβ6 Integrin Ligand Conjugates Reduce TGF*β*-Induced EMT in Human Non-Small Cell Lung Cancer

**DOI:** 10.3390/ijms24021475

**Published:** 2023-01-12

**Authors:** Elena Andreucci, Kelly Bugatti, Silvia Peppicelli, Jessica Ruzzolini, Matteo Lulli, Lido Calorini, Lucia Battistini, Franca Zanardi, Andrea Sartori, Francesca Bianchini

**Affiliations:** 1Department of Experimental and Clinical Biomedical Sciences “Mario Serio”, University of Florence, Viale Morgagni 50, 50134 Florence, Italy; 2Department of Food and Drug, University of Parma, Parco Area delle Scienze 27A, 43124 Parma, Italy

**Keywords:** A549 lung cancer cells, transforming growth factor*β* (TGF*β*), αvβ6 integrin ligands, epithelial-mesenchymal transition (EMT), nintedanib, molecular conjugates

## Abstract

Growth factors and cytokines released in the lung cancer microenvironment promote an epithelial-to-mesenchymal transition (EMT) that sustains the progression of neoplastic diseases. TGF*β* is one of the most powerful inducers of this transition, as it induces overexpression of the fibronectin receptor, αvβ6 integrin, in cancer cells which, in turn, is strongly associated with EMT. Thus, αvβ6 integrin receptors may be exploited as a target for the selective delivery of anti-tumor agents. We introduce three novel synthesized conjugates, in which a selective αvβ6 receptor ligand is linked to nintedanib, a potent kinase inhibitor used to treat advanced adenocarcinoma lung cancer in clinics. The αvβ6 integrin ligand directs nintedanib activity to the target cells of the tumor microenvironment, avoiding the onset of negative side effects in normal cells. We found that the three conjugates inhibit the adhesion of cancer cells to fibronectin in a concentration-dependent manner and that αvβ6-expressing cells internalized the conjugated compounds, thus permitting nintedanib to inhibit 2D and 3D cancer cell growth and suppress the clonogenic ability of the EMT phenotype as well as intervening in other aspects associated with the EMT transition. These results highlight αvβ6 receptors as privileged access points for dual-targeting molecular conjugates engaged in an efficient and precise strategy against non-small cell lung cancer.

## 1. Introduction

One of the main obstacles undermining the efficacy of anticancer therapies is the difficulty encountered in targeting cancerous cells inside the complex heterogeneous neoplastic tissue without affecting healthy tissue [1]. The tumor microenvironment (TME) is a complex entity where stromal and cancer cells, their secreted cytokines, growth factors, and bioactive mediators interact continuously [2,3,4]. This complex network strongly contributes to intensifying cancer progression. In particular, the transforming growth factor *β* (TGF*β*) is one of the most powerful inducers of the epithelial-to-mesenchymal transition (EMT) of cancer cells, a critical step in neoplastic progression [5]. Although the activation of this EMT phase is reversible, the persistence of a pro-inflammatory tumor microenvironment supports a constant slipping of cancer cells towards the mesenchymal phenotype, a degeneration of the situation, and the sparking of a vicious circle [6]. This phenotypic switch leads to a more invasive and aggressive phenotype and is often associated with the loss of epithelial markers, including E-cadherin and tight junction proteins [7,8], the increase in cancer cell proliferation and survival [9,10], and the development of drug resistance [11]. Moreover, besides steering the EMT towards the creation of cancer cells, TGF*β* also promotes the fibroblast-to-myofibroblast transition (FMT) and the endothelial-to-mesenchymal (EndoMT) transition programs [12,13,14,15]. In turn, these additional two processes increase the deposition of the extracellular matrix (ECM), enhance tissue matrix stiffness, and drive malignant progression [16,17].

Tremendous interest exists in identifying novel biomarkers of this mesenchymal switching, which may become targets for ‘cargoes’ of antitumoral drugs that can simultaneously counteract activation of stromal and cancer cells [18].

Lung cancer is the second most common cause of new cases of cancer worldwide and the leading cause of cancer deaths worldwide [19]. Lung cancer comprises two major histologic subtypes, small cell lung cancer (SCLC) and non-small cell lung cancer (NSCLC), with the latter representing the majority of lung tumors. Unfortunately, the majority of lung cancer diagnoses occur after the cancer has metastasized [20]. Although targeted therapy with tyrosine kinase inhibitors and chemotherapy provides patients with prolonged survival, the average survival rate of 5 years is still very low [21]. Thus, there is a pressing need for developing diverse and more efficient therapeutic approaches for patients with metastatic disease. 

Recently, nintedanib, a potent oral angiokinase inhibitor, that targets platelet-derived growth factor receptor- (PDGF-R), fibroblast growth factor receptor- (FGF-R), and vascular endothelial growth factor receptor- (VEGF-R) signaling, has proven effective and safe for NSCLC treatment in combination with docetaxel [22]. Interestingly, in patients with NSCLC, resistance to receptor tyrosine kinase inhibitor (RTKI) therapy depends on the activation of the EMT program [23] due principally to the release of activated TGF*β,* as previously mentioned [24,25]. Furthermore, EMT is closely associated with the overexpression of integrin receptors. Integrins are a superfamily of transmembrane heterodimeric adhesion receptors composed of alpha and beta subunits that regulate many of the cell–cell and cell–ECM interactions. Four different classes of integrin receptors have been described, depending on distinct domain structures recognized by these heterodimers. The five members of the αv integrin subfamily (αvβ1, αvβ3, αvβ5, αvβ6, and αvβ8) recognize the sequence arginine-glycine-aspartic acid (RGD) as the general binding motif. Interestingly, the RGD motif is present on different types of structural ECM proteins (e.g., vitronectin, fibronectin, and osteopontin), but also on other molecules of clinical interest [26,27]. Indeed, integrins that recognize the RGD motif, in particular the αvβ6 integrin, mediate the release and activation of TGFβ from the latency-associated peptide (LAP) in the extracellular storages following a protease-independent mechanism [28].

Several studies have shown that a high expression of RGD-recognizing integrins on cancer cells correlates with the acquisition of a more malignant phenotype characterized by increased cell migration, invasion, matrix metalloproteinase (MMP) secretion, and poor prognosis [29,30,31]. In NSCLC adenocarcinoma cells, TGF*β* induces the overexpression of the αvβ6 integrin receptor, and it has been observed that, in lung carcinomas, αvβ6 integrin promotes the adhesion of cancer cells to fibronectin, enhances tumorigenicity, and confers resistance to apoptosis induced by standard chemotherapeutic agents [32]. 

On these grounds, we sought to explore the tremendous potential of the αvβ6 integrin receptor that may be used either directly as a therapeutic target, or as a selective ‘hook’ for securing anti-tumoral drug delivery, via a molecular-precision approach. Thus, we envisioned overcoming the TGF*β*/EMT vicious circle by delivering an antitumor drug to the heterogeneous neoplastic lesions in a preclinical model of NSCLC.

Based on our previous findings [33,34], we synthesized three novel conjugates, compounds **1**–**3** (Figure 1), by linking nintedanib to an integrin ligand (IL) that selectively interacts with the αvβ6 integrin receptor. These conjugates differ in their type of linker (e.g., length and hydrophilicity), and in the number of IL units per molecule (monomeric **1** and **2** vs dimeric **3**). Interestingly, we already demonstrated that a similar class of conjugated compounds is active on αvβ6-expressing fibroblasts [35] and that the TKI activity of nintedanib is maintained even after covalent conjugation via its piperazine unit. In the present work, we found that two of the three synthesized molecules, namely compounds **1** and **2**, inhibit 2D and 3D cancer cell growth and reduce the clonogenic ability of TGF*β*-primed NSCLC cells. In addition, the treatment of TGF*β*-primed cancer cells with these compounds significantly reduces critical aspects of the acquired mesenchymal phenotype, severely compromising the TGF*β*-dependent EMT program [36,37].

## 2. Results

### 2.1. Synthesis and Characterization of Conjugated Compounds

The synthesis of novel conjugates **1**–**3** parallels that of similar nintedanib conjugates [35] and is described in the Supporting Information (Appendix A). The three compounds possess a molecule of nintedanib conjugated to an integrin targeting unit (IL)—an aminoproline-based cyclohexapeptidomimetic, *c*(AmpLRGDL) that has already proved efficient and selective in targeting the α_V_β_6_ integrin receptor [38]; they all share robust triazole-based and amide linkages. The linker in conjugate **2** has a glycolic portion that makes it more hydrophilic than conjugate **1**. Conjugate **3** possesses two IL units aimed at increasing binding to the integrin receptor and integrin-mediated cell internalization. Unconjugated ligand **4** was synthesized according to a previously reported procedure [38] and was used as a reference compound in biological assays. 

### 2.2. Integrin Expression

Expression of αvβ6 integrins was determined by cytofluorimetric analysis. Human NSCLC A549 and H1975 cells, either grown in standard conditions or exposed to TGFβ treatment, were detached from the substrate using Accutase^®^ to minimize the exposure of the extracellular membrane proteins to the enzymatic activity of trypsin. The cells were then incubated with primary antibodies, in the absence of membrane permeabilization. The cytofluorimetric procedures ensured quantification of the αvβ6 integrin receptor protruding outside the cells. We found that at least 34% of the A549 cell population and 11% of the H1975 cell population express the αvβ6 integrin receptor when grown in standard conditions (Figure 2A,C). These percentages increased to 66% of the parental population in A549 cells and to 55% in H1975, when the cells were exposed to TGF*β* for 48 h (Figure 2B,D). Exposure to TGFβ caused a significant decrease of αvβ3 integrin expression in A549 cells from 72% to 32% (Appendix A), whilst αvβ3 integrin expression in H1975 was unaffected by TGFβ treatment (98%, Appendix A). Cytofluorimetric analysis of K562 erythroleukemia cells revealed that they do not express either αvβ6 or αvβ3 integrin receptors (PMID: 35664627); therefore, these cells were used as a negative control (Appendix A). In addition, we evaluated αvβ6 integrin expression on human melanoma cells (SSM2c, Appendix A) and murine fibroblasts (L929, Appendix A), whereas expression of the αvβ3 integrin receptor in these cells had been previously evaluated [35,39]. 

### 2.3. Inhibition of Adhesion

To verify whether conjugates **1**–**3** were able to bind the integrin receptor efficiently and compete with the receptor’s natural ligand (fibronectin) for the binding site, we performed an inhibition of adhesion assay. In this assay, the biological effects of the three conjugated compounds were compared to the inhibitory ability of the unconjugated ligand **4**. Inhibition of A549 and H1975 cell adhesion to fibronectin was determined in both untreated and TGF*β*-treated cells. The assay was carried out with increasing concentrations of compounds **1**–**3** and unconjugated cyclic peptidomimetic **4**. As shown in Figure 3, all compounds inhibited cell adhesion in a concentration-related manner. In each case, the inhibition obtained in TGF*β*-treated cells was more pronounced than that seen in the untreated cell cultures. 

The highest levels of inhibition in A549 cells were registered with compounds **1** and **2**, which were able to inhibit more than 50% adhesion in TGF*β*-treated cells even at concentrations of 1 μM, a percentage that raised to 70% when these compounds were tested at 5 μM. Quite surprisingly, dimeric conjugate **3** proved to be the least-performing compound in the series. Finally, the inhibition obtained using compound **4** showed a weak concentration dependence, and even in TGF*β*-treated cells, the level of inhibition did not surpass 50% (Figure 3A,B). Moreover, high levels of inhibition were registered in H1975 cells with compounds **1**–**3** that reached a minimum of 50% inhibition of adhesion in TGF*β*-treated cells at 1 μM. These percentages increased to 70% when the compounds were tested at 5 μM. Interestingly, in H1975/TGF*β*-treated cells, dimeric conjugate **3** performed similarly to the other two compounds. Finally, the inhibition obtained using compound **4** in H1975/TGF*β*-treated cells, showed a weak concentration-dependence (Figure 3C,D). As already mentioned, the expression of αvβ3 integrins in A549 cells was reduced after exposure to TGF*β*. Since αvβ3 integrins are one of the most avid receptors for vitronectin (VN), we investigated whether compounds **1**–**4** were able to inhibit A549 cell adhesion to VN. We discovered that, in A549 cells and in TGF*β*-treated A549 cells, cell adhesion to vitronectin was not inhibited by the three different conjugated compounds nor by compound **4** (Appendix A); this lack of efficacy is probably due to the lower binding affinity of the integrin ligand towards the αvβ3 integrin receptors. Indeed, the binding affinity of *c*(AmpLRGDL) cyclohexapeptide **4** for isolated αvβ6 and αvβ3 integrins were previously determined and already known (IC_50(αvβ6)_ = 8.3 ± 0.4 nM, IC_50(αvβ3)_ = 2122 ± 266 nM, selectivity IC_50(αvβ3)_/IC_50(αvβ6)_ = 255) [38].

### 2.4. Cell Internalization of Conjugated Compounds **1**–**3** and Nintedanib

Since nintedanib binds to the intracellular portion of growth factor receptors, it is important to measure the ability of conjugates **1**–**3** to be internalized and the role of αvβ6 integrin in the internalization process. 

Cell internalization was assessed by both cytofluorimetric and confocal analysis, exploiting the intrinsic fluorescence of the nintedanib moiety (λ_exc_ = 405 nm, λ_em_ = 421 nm) [35]. 

A549 and H1975 cells, treated for 48h with TGFβ, were exposed for 24 h to the different compounds **1**–**3** or nintedanib at concentrations of 5 µM. After this, the percentage of cells that proved positive to compound internalization after each of the different treatments was determined and compared with the fluorescence intensity of untreated cells (Figure 4). About 99% of A549/TGFβ-treated cells exposed to nintedanib proved positive. In these cells, 74% of cells were positive after exposure to compound **1**, whilst after exposure to compounds **2** and **3,** positive cells ranged between 43.8 and 49% (Figure 4A). As expected, about 100% of H1975/TGFβ-treated cells that were exposed to nintedanib were positive. In these cells, 55.5% of cells were positive to compound **1**, 35% were positive to compound **2**, whilst surprisingly, compound **3** showed a very weak propensity towards internalization (Figure 4B). In addition, we investigated the internalization in αvβ6-negative K562 cells. We found the percentage of positive cells within the K562 population to be very high (95–100%) after incubation with free nintedanib, whereas the fluorescence associated with K562 cells exposed to compounds **1**–**3** was very weak, similar to that of untreated cells (Appendix A). Finally, two other αvβ6 integrin-expressing cell lines were used to evaluate the internalization of these compounds, namely the human melanoma cells SSM2c (Appendix A) and the L929 murine fibroblasts (Appendix A). Again, compounds **1** and **2** were better internalized than compound **3**, confirming the trend observed with NSCLC cells. Moreover, we added to our findings by investigating whether internalization of the compounds was affected by the concentrations involved, and we found that the percentages of the positive population, in both A549 cells and TGF*β*-treated A549 cells increased proportionally according to the treatment they were exposed to (Appendix A).

To confirm the intracellular localization, A549 cells and TGF*β*-treated A549 cells were also examined by confocal microscopy analysis after 24 h incubation with compounds **1**–**3** or nintedanib (5 μM). We observed an intense spot-like staining pattern in cells exposed to compounds **1** and **2**, probably associated with endocytic vesicles. Conversely, cells exposed to compound **3** express a faint granular fluorescent signal mostly distributed close to the cell nuclei. Interestingly, TGF*β*-treated cells showed a stronger granular staining pattern than that registered for untreated cells, confirming a higher degree of internalization for TGF*β*-activated cells, in complete accordance with the cytofluorimetric results (Figure 5).

### 2.5. Tyrosine Kinase Inhibition Activity 

Before investigating the effect of compounds **1**–**3** on cell proliferation, we evaluated whether the TK inhibition activity of nintedanib was retained after the covalent conjugation of the nintedanibpiperazine nucleus. Our previous data from analogous conjugates [35] suggested that the ability of the nintedanib unit to interact with growth factor receptors is not compromised by the functionalization involving the piperazine nitrogen. For confirmation, compound **1** was tested against human recombinant VEGFR2 (Eurofins Cerep laboratories), and it was found that its activity compared well with that of free nintedanib. Inhibition of receptor phosphorylation was measured by a radiometric detection method. IC_50_ values were determined from a nine-point concentration−response curve generated using the standard assay conditions for the target. Conjugated compound **1** and nintedanib exhibited IC_50_ values of 6 nM and 7 nM, respectively (Appendix A). Once the activity of the nintedanib moiety in our conjugates was verified, we proceeded in evaluating the inhibition of conjugates **1**–**3** and of nintedanib on the proliferation of TGF*β*-treated A549 cells by MTT assay. We found that compounds **1** and **2** were slightly less efficient at inhibiting cell proliferation (IC_50_ = 9.5 μM and 10.1 μM, respectively) than nintedanib. Indeed, proliferation of TGF*β*-treated A549 cells was almost completely inhibited by treatment with nintedanib at concentrations >10 μM (IC_50_ = 7.5 μM) (Appendix A).

### 2.6. Inhibition of α-Smooth Muscle Actin Protein Expression

Expression of α-smooth muscle actin (α-SMA), a well-known biomarker of myofibroblast differentiation, was used to distinguish which of the three different compounds was the most efficient in inhibiting TGFβ-driven EMT and hence the most likely to be chosen as the best dual-target compound. Inhibition of α-SMA protein expression in TGF*β*-treated A549 cells (untreated cells, Figure 6A) was evaluated after treatment with nintedanib, compounds **1**–**3**, integrin ligand **4**, or the combination of compound **4** with nintedanib for 24 h. We found that α-SMA expression was reduced by half after exposure to nintedanib alone or in combination with compound **4**. Compound **4** by itself was not able to affect α-SMA expression. Moreover, compounds **1** and **3** also did not affect α-SMA expression. Interestingly, compound **2** significantly reduced α-SMA expression in A549/TGFβ treated cells more efficiently than nintedanib alone and more efficiently than the combination nintedanib compound **4**. 

### 2.7. Uptake and Inhibition of Tumor Spheroid Growth and Migration

Heterogeneous spheroids of tumor cells and fibroblasts were generated using the hanging drop technique to mimic the complex three-dimensional environment of tumor tissues. Spheroids were exposed 24 h to nintedanib or to different compounds at concentrations of 5 µM for 24 h. Next, the fluorescence intensities associated with each cell population were analyzed as previously described. The highest fluorescence intensity was associated with the tumor spheroids exposed to nintedanib (92.5%) (Figure 7A). The fluorescence intensities of the tumor spheroids exposed to compounds **1**, **2**, and **3** were 71.2%, 65.7%, and 47.2%, respectively. Tumor spheroids were next exposed to nintedanib, or compounds **1**, **2** or integrin ligand **4,** or to the combination of compound **4** with nintedanib for 72 h, and the 3D growth was quantified by comparing the increase in spheroid diameter with that at time 0 and normalized to untreated spheroids. We found that spheroid growth was inhibited significantly after treatment with nintedanib, compounds **1**, **2** and the combination of compound **4** with nintedanib, whilst integrin ligand **4** alone did not induce a significant reduction of 3D growth (Figure 7B,C). In addition, cell spheroids at 72 h time intervals maintained intracellular fluorescence (Figure 7D). Interestingly, we observed similar behavior in spheroids obtained with L929 cells exposed to nintedanib, compound **1**, or compound **2** (Appendix A). 

Heterogeneous spheroids of tumor cells and fibroblasts were also used to evaluate the ability of the compounds to inhibit 2D flat surface migration and growth once the spheroids were allowed to attach to a suitable surface (Figure 8). We found that nintedanib and compound **2**, as well as compound **4** in association with nintedanib, significantly impaired both 2D migration and 2D growth, whereas compound **1** inhibited cell spreading and migration to a lesser extent. Free ligand **4** did not inhibit cell growth and migration, behaving similarly to untreated cells. 

### 2.8. Inhibition of the Epithelial-Mesenchymal Transition 

The ability of compounds **1** and **2** to inhibit the EMT in TGF*β*-treated A549 cells was evaluated, in comparison to nintedanib, using various in vitro experiments. The first assay envisaged the evaluation of cellular invasiveness of TGF*β*-treated A549 cells by measuring the capacity of cells to migrate through fibronectin-coated filters. Compounds **1** and **2** reduced the invasiveness of TGF*β*-treated A549 cells pleasingly, even if not as efficiently as nintedanib, which exhibited 80% migration inhibition with respect to the control experiments (Figure 9A). 

The second piece of evidence was obtained by confocal analysis. We found, as expected, that cells exposed to TGF*β* showed a more complex F-actin bundle network compared to untreated cells. Interestingly, the F-actin bundles were partially disassembled in cells exposed to TGFβ and compounds **1** or **2**, as highlighted by the presence of diffused dots; however, a more significant reduction in the assembly of F-actin filaments was witnessed upon exposure to TGF*β* and nintedanib (Figure 9B,C). We next analyzed the expression of E-Cadherin (E-Cad) and found it to be enhanced after treatment with nintedanib. Interestingly, cells exposed to both compounds **1** and **2** showed a higher expression of E-Cad compared to that of TGFβ-treated cells (Figure 9D).

### 2.9. Inhibition of Clonogenic Activity

Clonogenic assays were performed on selected TGFβ-treated A549 cells that gave a high fluorescent signal upon treatment with nintedanib, compounds **1** or **2**, indicating a high level of drug internalization. The selected cells were seeded at low density and allowed to grow in colonies; their ability to form colonies was compared to that of TGF*β*-treated A549 cells that were not previously exposed to nintedanib or conjugates. We found that untreated cells showed a high rate of colony formation, which was significantly reduced by the treatment with nintedanib (50%). Interestingly, conjugates **1** and **2** were definitely more efficient than free nintedanib in impacting colony formation (70% reduction, as compared to untreated cells) (Figure 10B); in fact, treatments with compounds **1** or **2** induced a significant reduction in both number and size of colonies (Figure 10C).

## 3. Discussion

Activation of the EMT program is a transitory phenomenon sustained by cytokines, growth factors, and other pro-tumor inducers released in the TME by cancer and stroma cells. Different types of cells and mediators come together in a vicious circle which fuels the EMT program. In NSCLC, the activation of the EMT program has been associated with epithelial growth factor receptor (EGFR)-TKI resistance [40], and different strategies have been devised to interfere with TGF𝛽-induced EMT. In this context, different clinical trials targeting TGF𝛽-involving pathways have been conducted [41]. One of the consequences of EMT onset in lung adenocarcinoma cells is the overexpression of αv𝛽6 integrin receptors, which plays a fundamental role in the activation of TGF𝛽; indeed, αv𝛽6 integrin expression is strongly associated with poor prognosis in lung cancer [42].

In previous experimental models of human cancers, we demonstrated that certain dual conjugates, constructed by αv𝛽3 integrin ligands covalently linked to the RTK inhibitor sunitinib, preferentially accumulated at the tumor site and proved more efficient in inhibiting in vivo tumor growth than unconjugated sunitinib [33,34]. Along these lines, we introduced three novel compounds, obtained by the covalent conjugation of a selective αv𝛽6 integrin ligand [38] with nintedanib, an RTK inhibitor currently used in lung cancer treatment [22]. 

The three different conjugates, compounds **1**–**3**, were prepared by joining a nintedanib moiety with the integrin ligand by using three different triazole-bearing uncleavable linkers featuring either an alkyl chain (compound **1**), a short PEG chain (compound **2**), or a PEGylated dimeric ligand presentation (compound **3**) (Figure 1). We investigated the effect of the three different compounds on human advanced lung adenocarcinoma cell lines. We chose the A549 cell line, representative of EGFR wild-type, and the H1975 cell line, representative of NSCLC with EGFR T790M mutation, resistant to EGFR-TKI [43]. Firstly, the expression of αv𝛽6 integrin receptors was determined in A549 and H1975 cells exposed to standard in vitro conditions (Figure 2A and Figure 2C, respectively), or in cells exposed to TGFβ treatment (Figure 2B,D), revealing that TGFβ treatment induced an increase in αv𝛽6 integrin expression in both cell lines. 

Next, we verified the propensity of both the integrin ligand moiety and the nintedanib moiety, united in the covalent presentation, towards their respective biological targets. The RTK signaling inhibitory activity of compound **1**, taken as a representative example of the three conjugates, was carried out by certified enzyme profiling services, which confirmed an estimated IC_50_ of 6 nM for compound **1** against VEGFR phosphorylation (Kinase Profiler™, IC 50 Profiler™ Eurofins, Celle-Lévescault France), in line with that of free nintedanib (Appendix A).

The ability of conjugates **1**–**3** to interact with the αv𝛽6 receptor was evaluated using the inhibition of adhesion assay. We observed that conjugates **1**–**3** inhibit cancer cell adhesion to FN in a concentration-dependent manner in both A549 and H1975 cells grown in standard conditions (Figure 3A and Figure 3C, respectively). The percentages of inhibition were similar to those obtained using the unconjugated compound **4**. In TGF*β*-treated cells, we found a stronger concentration-dependent inhibition of adhesion, probably due to the higher expression of the αv𝛽6 receptor. Interestingly, the percentages of inhibition in A549/TGF*β*-treated cells were significantly higher when cells were exposed to conjugates **1** and **2** compared to compound **4** (Figure 3B), whilst in H1975/TGF*β*-treated cells, the percentages of inhibition of adhesion were significantly higher when the cells were exposed to the three conjugates compared to compound **4** (Figure 3D). The concentration-dependent pattern confirmed that the unconjugated ligand **4** and the three conjugated compounds interact with the FN receptor in cells grown under standard conditions. However, even more interesting results were obtained when A549 and H1975 cells were exposed to TGF*β* treatment, which induced an increase in the expression of the integrin receptor. Not only did we observe a significant difference in the inhibitory ability between the three conjugates and the unconjugated ligand, but a difference in the inhibitory behavior of the conjugated compounds in the two cell types was also seen. This different behavior may be explained by the different overall structures of the conjugates, and hence their distinctive physicochemical properties. In addition, the different behavior in the two cell lines may be explained by considering that the ligand–receptor interaction, on an extracellular level, depends on many factors, including differences in membrane fluidity, glycocalyx composition, receptor recycling, and clustering ability. 

One of the major challenges in using the αv-subclass of the integrin receptors as targets for precision medicine is that, although different cell types present a characteristic αv expression profile, αv integrins ultimately recognize the same peptide motif. To overcome this issue, we indirectly evaluated the selectivity of the ligand–integrin receptor interaction, taking into account the ability of the ligands to inhibit the adhesion of the αvβ3 integrin to the vitronectin substrata. Indeed, we found that A549 expressed low levels of αv𝛽3 integrin compared to αv𝛽6; moreover, the αv𝛽3 expression was downregulated after TGF𝛽 exposure (Appendix A). To explain this behavior, we suggest that TGF𝛽 promotes a preferential expression of the 𝛽6 subunit over the 𝛽3 subunit, in analogy to the integrin switch observed in other cancer types [44,45]. The downregulation of αvβ3, in association with overexpression of αvβ6, may depend on a ‘hierarchy of preferential dimerization’ where αv prefers to heterodimerize with different β subunits, leading to the reduction of the expression of one integrin in favor of the other. In line with these observations, poor inhibition of A549 adhesion to VN, an αv𝛽3 endogenous ligand, was found (Appendix A). According to our previous results on sunitinib-conjugates, [33,34], we expected the dimeric compound **3**, bearing two integrin ligands per molecule, to be more efficient in inhibiting cell adhesion than the monomeric counterparts **1** and **2**, but this was not the case since compounds **1** and **2** performed better than **3** (Figure 3). It is possible that the actual arrangement of the integrin-recognizing peptide moieties within dimer **3** does not fulfil the spatial requisites (i.e., density and distance) of the surface-exposed αv𝛽6 integrin receptors and therefore, no enhancement of receptor recognition is achieved.

The evaluation of ligand–receptor interaction and subsequent cell uptake was conveniently carried out by exploiting the intrinsic fluorescent properties of the nintedanib portion. Cytofluorimetric assay confirmed that compounds **1** and **2** are internalized more efficiently than compound **3** (Figure 4), and their intracellular localization was also ascertained by confocal analysis (Figure 5). Selective integrin-mediated internalization of the conjugates was supported by the fact that compounds **1**–**3** were not internalized in K562 cells that did not express αv𝛽6 integrins. Importantly, while free nintedanib unselectively passed the cell membranes of all the cells tested, conjugates **1**–**3** were able to selectively accumulate in those cells that overexpress αv𝛽6 integrins. Moreover, TGF𝛽-priming of cancer cells determined an increase of cell uptake of the conjugates due to higher expression of the αv𝛽6 integrin receptors.

To investigate whether the compounds were able to reduce cell viability, we evaluated the inhibition of cell proliferation compared to free nintedanib. We found that both nintedanib and the three conjugates showed a concentration-dependent inhibition of cell proliferation. In particular, at high concentrations (50 μM), the inhibition of proliferation by compound **2** and nintedanib were comparable, while compound **3** was the least efficient of the series (Appendix A).

Next, we explored the ability of compounds **1**–**3** in triggering EMT reversion in TGF𝛽-primed cells. It is worth underlying that these studies were performed at a sub-lethal concentration of the drugs to analyze their functions in a population of viable cells. Compound concentrations of 5 μM for these assays were judged convenient and compatible with good levels of internalization and inhibition of proliferation (vide supra). Firstly, we investigated the ability of the conjugates to inhibit α-SMA expression, a well-known marker of the mesenchymal phenotype [46]. Expression of α-SMA was evaluated in cells exposed to the different treatments, namely equimolar concentration of nintedanib and the three conjugates, as well as the combination of free nintedanib and the unconjugated integrin ligand **4**, in order to appreciate the impact of the two active moieties—the integrin ligand and nintedanib drug—when delivered in combination or under a covalent conjunction.

In these assays, nintedanib inhibited α-SMA expression by half compared to untreated cells. Compounds **1**, **3**, and **4** were not effective in inhibiting α-SMA expression, whereas conjugate **2** and the nintedanib + **4** combination significantly inhibited α-SMA expression (Figure 6). Interestingly, compound **2** performed even more efficiently than the nintedanib + **4** combination, despite being internalized to a lesser extent compared to nintedanib.

In previous studies, free RGD-based ligands proved to partially inhibit EMT; however, in our experimental protocol, the free αv𝛽6 integrin ligand (compound **4**) was not sufficient to reduce α-SMA expression [47,48]. As a hypothesis, an independent pathway engaging the αv𝛽6 integrin and the entire molecule of compound **2** may have a role in α-SMA regulation, but additional investigation is required to sustain this supposition. Indeed, while nintedanib inhibits the effect of exogenous TGF, conjugated compound **2** could also play a role, better than the other compounds, in the autocrine αv𝛽6-mediated activation of TGF𝛽 from extracellular storages. These results confirm that understanding the interactions between growth factors and integrin signaling is extremely complex and requires further investigation [49]. 

Subsequently, to investigate in-depth the effects of conjugated compounds on EMT reversion in TGF𝛽-treated cells, we extended our protocol to a 3D model (Figure 7). To better recapitulate the heterogeneity of the adenocarcinoma tumor lesion, we employed tumor spheroids formed by the combination of two different cell types; as a proof of concept, 3D spheroids obtained from a 1:1 mixture of murine fibroblasts and A549 cells (primed with TGF𝛽) were used. Cellular internalization is known to be affected by cell density and tissue stiffness [50,51]. In this regard, we demonstrated that the conjugated compounds enter the tumor spheroids cells and efficiently inhibit their growth and migration. In particular, the fluorescence associated with tumor spheroids exposed to compound **3** for 24 h was lower than that of spheroids exposed to the other compounds or nintedanib. Moreover, we found that compounds **1** and **2** reduced the growth of tumor spheroids in a similar way to nintedanib treatment. Exposure to compound **4** determined just a transient reduction in tumor spheroid growth; indeed, after 48 h, spheroids recommenced their 3D expansion. No significant improvement in the inhibition of 3D growth occurred when the spheroids were exposed to the nintedanib + **4** combination. Interestingly, at the end of the assay (72 h), compounds **1** and **2** remained internalized in the tumor spheroids.

We next evaluated the inhibition of cell migration from 3D spheroids in a 2D flat migration assay. In this experimental procedure, the ability of the assayed compounds to inhibit spheroid growth was also evaluated. Interestingly, we found a strong inhibition of collective migration when spheroids were exposed to nintedanib, compound **2**, or nintedanib + **4**, despite synergistic effect not being witnessed. Compounds **1** and **4** exerted a considerably weaker effect on spheroid migration, in comparison to that of nintedanib. In terms of inhibition of spheroid core growth, in this 2D model, the effect of compound **2** was similar to that of nintedanib alone or in combination with compound **4**, while treatment of spheroids with compound **1** or **4** determined a very weak inhibition of the core growth. Collectively, these results confirm that, among the three different conjugated compounds, compound **3** was the least active, and no additive/synergistic effect was observed by the co-administration of nintedanib and compound **4**. We then proceeded to compare the effect of nintedanib to that of the promising compounds **1** and **2**. 

The actin cytoskeletal rearrangement is a critical feature of cells engaged in EMT. TGF𝛽 treatment increased F-actin cytoskeletal organization in A549 cells, and we found that compounds **1** and **2** partially disassembled the cytoskeletal network (Figure 9). In previous reports, it was found that, upon EMT, the tight junctions between cancer cells dismantle and favor the migration of NSCLC cells through a fibronectin-rich extracellular matrix [52]. Along these lines, we found that E-cadherin, a marker of EMT, was re-expressed on A549- and TGF𝛽-treated A549-cells after exposure to compounds **1** and **2**. The acquisition of an EMT phenotype correlates with a more aggressive ability of the cells to invade surrounding tissue; indeed, compounds **1** and **2** significantly reduced invasiveness of TGF𝛽-treated A549 cells. Taken together, the effect of nintedanib was similar and even stronger to that of compounds **1** and **2**, meaning that we did not find a cooperative effect of the two active units (integrin ligand and nintedanib). As mentioned before, we hypothesize that these results depend on the ability of nintedanib to freely enter the cells, while conjugated compounds need a receptor-dependent internalization process. During the invasive process, the adhesive machinery is important, but the downstream inhibitory signal, exerted by the RTKI, is also crucial. It has been observed that, depending on the involvement or not of mechanical tension, the integrin–ligand interaction will produce functional receptor crosslinking and thus the ligand–receptor binding triggers migration movement and downstream signal transduction [52,53]. Indeed, many different downstream signaling pathways are activated by mechanical signals from ECM and transmitted through integrins. Thus, the effect of conjugated compounds, which enter the cells to a lesser extent than nintedanib, only partially reduces tumor cell invasiveness.

Considering that this study focuses on the opportunity of using a dual-conjugated drug to target cells that overexpress αv𝛽6 integrins, we investigated the ability of compounds **1** and **2** to reduce colony formation of TGF𝛽-treated cells as an in vitro model of the development of metastatic niche formation. We found that nintedanib reduced the number of colonies, but compounds **1** and **2** were significantly more efficient in doing so. These results could be explained by considering that the final effect on colony formation depends on both the amount of drug that has been internalized and the direct effect of nintedanib on adhesion to substrata. Moreover, free nintedanib might be sequestered inside lysosomal compartments as a mechanism of resistance, as clearly indicated by the presence of spotted signals in immunofluorescence assay [54], while conjugated compounds **1** and **2** could represent a valid alternative of escape from this lysosomal sequestration.

In conclusion, we have demonstrated that the phenotypic switch induced by TGF𝛽 in NSCLC, which leads to the acquisition of a more invasive and aggressive phenotype and is associated with αv𝛽6 overexpression, is reversed by the treatment with dual conjugated drugs of type **1** and **2** that target RTK pathways in αv𝛽6 integrin-expressing cells. Recently, the therapeutic scenario of NSCLC has changed substantially, and the need for a multidisciplinary approach has become progressively urgent. In particular, decisions in treatment depend on a precise histological and biological characterization of small biopsies, and often require challenging in-tissue sampling at diagnosis. The identification of specific oncogenic alterations and other druggable targets, such as αvβ6 integrin receptors, might favor a more precise intervention and improve patient outcome.

Given the large incidence of lung cancer in the population, it is important to concentrate all possible efforts on understanding how to prevent, diagnose, and treat lung cancer. The progression in treatment of patients with unresectable NSCLC relies on the implementation of targeted therapies and immunotherapies, either alone or in combination with chemotherapy. Still, a considerable percentage of NSCLC patients may not be eligible for either targeted therapies or immune checkpoint inhibitors. Therefore, novel therapeutic targets are required to provide these patients with enhanced treatment options. [55]. In this scenario, the use of dual target therapy using an αv𝛽6 integrin ligand conjugated with RTKI may assume a fundamental role. The first benefit is the opportunity of targeting each type of αv𝛽6-expressing cell in the tumor microenvironment, namely both cancer and stroma cells such as cancer-associated fibroblasts. Moreover, the αv𝛽6 ligand may disarticulate the effect of TGF and its activation, which is known to lead to EMT, fibrosis, and immunosuppression.

We are aware that the data presented in the present paper, using different types of cells in 2D and 3D culture conditions, may be confirmed with in vivo studies. In addition, although we have shown that the conjugate is capable of inhibiting TGF𝛽-driven EMT, the ability of the αv𝛽6 ligand to inhibit the release of TGF𝛽 from extracellular storage cells should also be investigated. However, this latter investigation requires an in-depth study that goes beyond the scope of this work.

## 4. Materials and Methods

### 4.1. Synthesis and Characterization of Conjugated Compounds

The synthesis of conjugates **1**–**3** is described in the Supporting Information (Appendix A). Nintedanib (BIBF 1120) was purchased from MedChemExpress (D.B.A., Milano, Italia). The compounds were dissolved in DMSO at 10 mM concentration and diluted at least 1000 times using PBS buffer (ECB4004L, Dulbecco’s Phosphate Buffer Saline w/o Calcium w/o Magnesium, Euroclone, Milano, Italy).

### 4.2. Cell Cultures 

The human lung adenocarcinoma cell lines (A549 and H1975), the murine L929 fibroblast cell line, and the human erythroleukemia cell line K562 were purchased from the American Type Culture Collection (ATCC, Rockville, MD, USA). The human melanoma SSM2c cell line was kindly provided by Dr Barbara Stecca [56]. A549, H1975, SSM2c, and L929 cells were grown in Dulbecco’s modified Eagle medium, containing 4500 mg/L glucose (DMEM 4500, GIBCO, Thermo Fisher Scientific, Waltham, MA USA) supplemented with 10% fetal calf serum (FCS) at 37 °C in a humidified incubator containing 10% CO_2_. 1.0 × 10^6^ cells were seeded in 100 mm Sarstedt dishes and propagated every 3 days by incubation with a trypsin–EDTA solution. The human erythroleukemic cell line K562 was maintained at 37 °C in RPMI-1640 medium (GIBCO, Thermo Fisher Scientific, Waltham, MA USA) supplemented with 10% FCS in T25 culture flasks (Sarstedt, Milano, Italy) in humidified incubator with 5% CO_2_. For 3D co-cultures, the L929 murine fibroblast cell line was used as a proof of concept. L929 cells were grown in DMEM 4500 supplemented with 10% FCS in the same culture conditions. Cell cultures were periodically monitored for mycoplasma contamination using Chen’s fluorochrome test. 

### 4.3. Determination of αvβ6 Expression

A549, L929, and K562 cells were allowed to grow in standard conditions; in several experiments, A549 cells were grown in the presence or in the absence of recombinant human TGFβ (10 ng/mL) (#11343160 Immunotools Friesoythe Germany). After 48 h incubation, cells were collected using Accutase^®^ solution and resuspended in PBS with 1% BSA (A9418 Sigma-Aldrich Merk Life Science S.r.l. Milano, Italy) (cytofluorimetry buffer). After blocking, cells were incubated with 1 μg/20 μL buffer of anti-αvβ6 (Bioss bs 5791R, Woburn, MA, USA, as suggested dilution 1:20) (anti-human/mouse/rat) for 1 h at 4 °C, followed by 1 h incubation with FITC-conjugated anti-rabbit (24549933) secondary antibodies at a 1 μL/50 μL dilution [57] from Immunotools (Friesoythe, Germany), or with 5 μL (1 μg) of anti-αvβ3-FITC conjugated antibody in 20 μL buffer (Thermofisher, Waltham, MA USA 11-0519-42, as suggested 5 μL/test). After extensive washes using cytofluorimetric buffer, cells were analyzed using a BD FACSCanto™ flow cytometry system (BD Bioscience, Eysins, Switzerland).

### 4.4. Inhibition of Cell Adhesion to Fibronectin

96-well plates were coated with fibronectin (10 µg/mL) (sc-29011 SantaCruz, D.B.A., Milano, Italia) by overnight incubation at 4 °C. Plates were washed with Phosphate Buffered Saline (PBS) solution and then incubated at 37 °C for 1 h with PBS containing 1% Bovine Serum Albumin (BSA). A549 cells were allowed to grow for 24 h in the presence or in the absence of TGFβ (10 ng/mL). Next, A549 cells were washed by centrifugation with PBS, counted, and suspended in serum-free medium at 0.6 × 10^6^ cells/mL. Lung carcinoma cell suspensions were exposed to different amounts of compounds **1**–**3** (final concentration ranged from 10 µM to 50 nM). A pre-incubation at 37 °C for 30 min was performed to allow the ligand–receptor equilibrium to be reached. Assays were performed in the presence of 2 mmol/L MnCl_2_. Cells were then plated on FN substrata (5–6 × 10^4^ cells/well) (F4759, Sigma-Aldrich Merk Life Science S.r.l., Milano, Italy) and incubated at 37 °C for 1 h. After incubation, plates were washed with PBS to remove the non-adherent cells, and 200 µL of 0.5% crystal violet solution in 20% methanol (34860, Merk Life Science S.r.l., Milano, Italy) were added. After 2 h of incubation at 4 °C, cells were permeabilized with 50 µL of DMSO (C6164, Sigma-Aldrich Merk Life Science S.r.l. Milano, Italy) and plates were examined at 550 nm in a counter BioRad (550 Microplate Reader, BioRad, Milano, Italy) [34,35]. Experiments were conducted in triplicate and repeated at least three times. The values are expressed as % inhibition ± SEM of cell adhesion relative to untreated cells.

### 4.5. Evaluation of Cell Internalization of Nintedanib and Conjugated Compounds 1–3

Internalization of nintedanib was determined by exploiting the fluorescent emission of nintedanib by means of a cytofluorimetric assay [35]. Briefly, A549 cells were seeded in standard medium (2 × 10^6^ cells/100 mm Petri dishes, Sarstedt), and after adhesion, cells were grown in the presence or in the absence of TGFβ (10 ng/mL) for 48 h. Next cells were exposed to nintedanib or conjugated compounds at different concentrations. After 24 h, cells were gently detached using Accutase (ECB3056D, Euroclone, Pero, Milano, Italy) or trypsin and the fluorescence, associated with different cell populations, was evaluated using the BV480 channel of the FACSCanto BD instrument (using 405 nm laser). Untreated cells were used as the negative control. The same procedure was applied to evaluate the internalization in K562, SSM2c, and L929 cells. 

### 4.6. MTT Assay

A549 cell viability was assessed using MTT (3-(4,5-dimethylthiazol-2-yl)-2,5-diphenyltetrazolium bromide) tetrazolium reduction assay (M5655, Sigma Aldrich, Milan, Italy), as described in [58]. Briefly, A549 cells were treated for 48 h in complete medium with TGFβ (10 ng/mL). Next, cells (2.5 × 10^3^) were plated into 96-multiwell plates in 100 µL of complete medium. After adhesion, the compounds were added to the medium at different concentrations. After 24 h exposure, the medium was replaced with 100 µL MTT reagent in each well, and plates were incubated at 37 °C. After 2 h, MTT was removed, and the blue MTT–formazan product was solubilized with dimethyl sulfoxide (DMSO, Sigma Aldrich). The absorbance of the formazan solution was read at 595 nm using the microplate reader (Bio-Rad, Milan, Italy). Inhibition of cell proliferation was calculated by the IC_50_ calculator (AAT Bioquest, Inc., Pleasanton, CA, USA, https://www.aatbio.com/ (accessed on 1 July 2022)).

### 4.7. Western Blot Assay

Western blot analysis was performed on whole-cell lysates of A549 cells, exposed for 48 h to TGF*β* treatment and subsequently exposed to 5 µM concentrations of different compounds. After 24 h incubation, media were rapidly removed and cell monolayers were lysed as previously described in RIPA lysis buffer, with the addition of proteinase inhibitor cocktail (04693124001 Roche Merk Life Science S.r.l., Milano, Italy) for 30 min in ice [35]. Protein content was quantified, and forty-fifty micrograms of total proteins were separated on Bolt^®^ Bis-Tris Plus gels 4–12% (Life Technologies, Monza, Italy). Fractionated proteins were transferred from the gel to a PVDF nitrocellulose membrane using iBlot 2 system (Life Technologies, Monza, Italy). Membranes were blocked for 5 min at room temperature, with EveryBlot Biorad blocking buffer (Life Technologies, Monza, Italy). Subsequently, the membranes were probed at 4 °C overnight with appropriate primary antibodies diluted in a solution of 1:1 Odyssey blocking buffer/T-PBS buffer, washed four times with PBS-Tween 0.1% solution, and probed with the secondary IRDye antibodies according to manufacturering instructions. The protein bands were analyzed by the Odyssey Infrared Imaging System (Lycor Bioscience) using software for protein quantification. The primary antibodies were mouse anti-alpha smooth muscle actin (human, mouse rat) (1:1000, Genetex GTX100034 Alton Pkwy, Irvine, CA, USA), rabbit anti-E Cadherin (1:1000, Cell signaling Technology, Danvers, MA USA), and mouse anti-tubulin (1:1000, Cell signaling Technology) that was used to assess the equal amounts of protein loaded in each lane.

### 4.8. Inhibition of Multicellular Tumor Spheroids (MCTS) Migration and Growth

A549 cells, exposed for 48 h to TGF*β* treatment, were mixed in a 1:1 ratio with murine L929 fibroblast (40,000/mL) in order to obtain 3D multicellular tumor spheroids (MCTS) resembling cellular heterogenicity of the neoplastic lesion. We found that L929 cells in a 1:1 ratio sustains the growth and survival of MCTS better than Matrigel. The spheroids were less apoptotic and more resistant to drug-induced apoptosis. We found that 25 µL of cell suspension were allowed to aggregate according to the hanging drop method [59]. After one week, multicellular tumor spheroids were completely formed. Tumor spheroids were next used in different experimental procedures. We evaluated 2D flat surface inhibition of migration and growth, and the inhibition of 3D growth. For 2D flat surface migration and growth inhibition assays, the spheroids were placed on regular culture dishes and allowed to spread and grow in complete medium containing different compounds at 5 µM concentration. The areas of cell spreading were used as parameters for migration (red lines), while the areas of the spheroid core were used as parameters for 2D growth (yellow lines). For 3D growth inhibition, spheroids were placed on 2% agarose coated dishes in complete medium containing the different treatments at 5 µM concentration. Spheroid areas were analysed and compared to time 0 and used to evaluate the inhibition of 3D growth (yellow lines). Images of all spheroids were taken daily using the EVOS XL Core Imaging System with a 10x objective (Thermofisher Scientific, Waltham, MA, USA ). All images were analysed using the open-source software ImageJ (1.52a, National Institutes of Health, Bethesda, MD, USA).

### 4.9. Immunofluorescence

A549 cells, exposed for 48 h to TGFβ treatment, were grown on fibronectin-coated (10 µg/mL) (sc-29011 SantaCruz, D.B.A., Milano Italy) glass coverslips in 6-well plates for 2 h, after which adherent cells were exposed to the different treatments, in the absence of FCS, for 24h. After incubation, the culture medium was removed, cells were washed with PBS, and fixed 30 min at 4 °C with 3.7% paraformaldehyde. Cells were permeabilized for 15 min with PBS 0.1% Tryton X-100 at room temperature, and then incubated 1 h at room temperature with blocking buffer (0.1% Tryton X-100 and 5.5% horse serum PBS). Next, cells were exposed to a 1 h-incubation at room temperature with rhodamine-phalloidin (Invitrogen Molecular Probes R415 Invitrogen AG, Basel, Switzerland), and incubation was continued for 45 min at room temperature in the dark. Subsequently, cells were dried, mounted onto glass slides, and examined with confocal microscopy using Leica SP8 confocal microscope. A single composite image was obtained by superimposition of 20 optical sections for each sample analyzed.

### 4.10. Invasion Assay

Invasiveness of A549 cells was determined in vitro on fibronectin-precoated polycarbonate inserts (Sarstedt 83.3932.800, 8µm pore size, 0.3 cm^2^ growth surface). Briefly, 50 µL of 10 μg/mL of fibronectin solution were seeded on each insert and allowed to dry in sterile conditions (1.6 µg/cm^2^). A549 human lung carcinoma cells were grown in the presence of TGFβ for 48 h. After incubation, 3.5 × 10^4^ cells were exposed to the different compounds, in serum-free medium (at 5 µM concentration), seeded in the upper compartment of the insert, and were free to perform in the absence of any chemoattractant. Inserts were incubated for 24 h at 37 °C in 10% CO_2_ in air. After incubation, the non-invading cells on the upper surface of the insert were wiped off mechanically with a cotton swab, while migrated cells on the lower side of the inserts were fixed in ice-cold methanol for 20 min, and then stained using the Diff-Quick kit (BD Biosciences, Eysins Switzerland). Photographs of randomly chosen fields were taken, and migrated cells were counted in 4 field/photograph.

### 4.11. Colonies

A549 lung adenocarcinoma cells were grown in the presence of TGFβ for 48 h. Cells were next exposed for 24 h to nintedanib or compound 1 or 2 at 5 μM concentration. At the end of the incubation, cells were sorted using a fluorescence activated cell sorter (Melody, Becton Dickinson, Bioscience, Eysins, Switzerland). Four different populations were isolated (TGFβ-treated A549 negative cells, TGFβ-treated A549 nintedanib positive cells, TGFβ-treated A549 compound 1-positive cells and TGFβ-treated A549 compound 2-positive cells). Activation of the labelled cells was provided by the violet laser (ex = 405 nm) while sorting of the activated cells was performed by their emitted wavelength at 480 nm (detection filter 450/50, using the BV480 channel). Isolated cells (1 × 10^3^) were seeded in a FN-coated 6 well dishes. These isolated cells were cultured for 2 weeks. The medium was changed every 4 days until the emergence of colonies. The colonies were fixed and stained with Diff-Quick solution.

### 4.12. Statistics

All data, presented as mean ± SEM, were obtained from at least three independent experiments (*n* = 3). Statistical analysis between two groups was performed using unpaired Student’s t-tests. In the case of comparative analysis of three or more groups, one-way analysis of variance (ANOVA) was performed followed by Tukey’s post hoc test. *p*-values were calculated using GraphPad Prism 6 and are provided in the figure legends. Band intensities in Western blot analysis were quantified using computer-based ImageJ software.

## Figures and Tables

**Figure 1 ijms-24-01475-f001:**
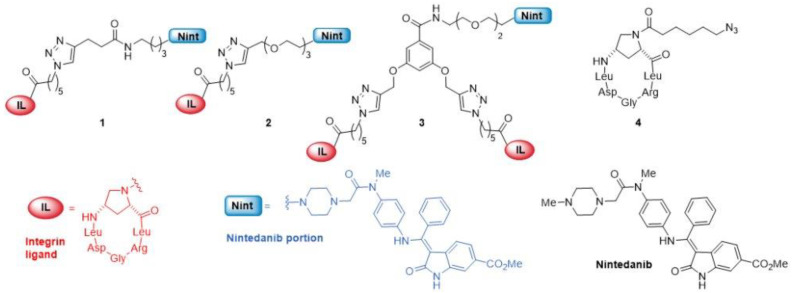
The three-integrin ligand (IL)-nintedanib (Nint) conjugates **1**–**3** of this work. The IL is a previously reported cyclic peptidomimetic [*c*(AmpLRGDL)] incorporating the integrin-recognizing sequence RGD and the *cis*-4-amino-L-proline residue (Amp). Unconjugated ligand **4** and nintedanib were used as reference compounds in the biological assays.

**Figure 2 ijms-24-01475-f002:**
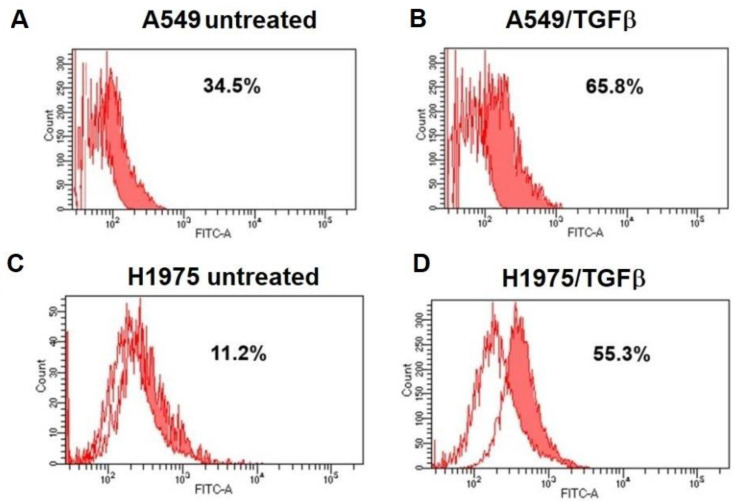
Expression of 𝛼v𝛽6 integrin receptor in A549 (**A**,**B**) and H1975 (**C**,**D**) NSCLC cells. Cells were grown for 48 h in the absence (left panels, **A**,**C**) or in the presence (right panels, **B**,**D**) of TGF𝛽 (10 ng/mL). Cells were stained with mouse anti-human αvβ6 antibody followed by FITC-conjugated anti-mouse immunoglobulin (full histograms). As a negative control (open histogram), cells were stained with FITC-conjugated anti-mouse immunoglobulin.

**Figure 3 ijms-24-01475-f003:**
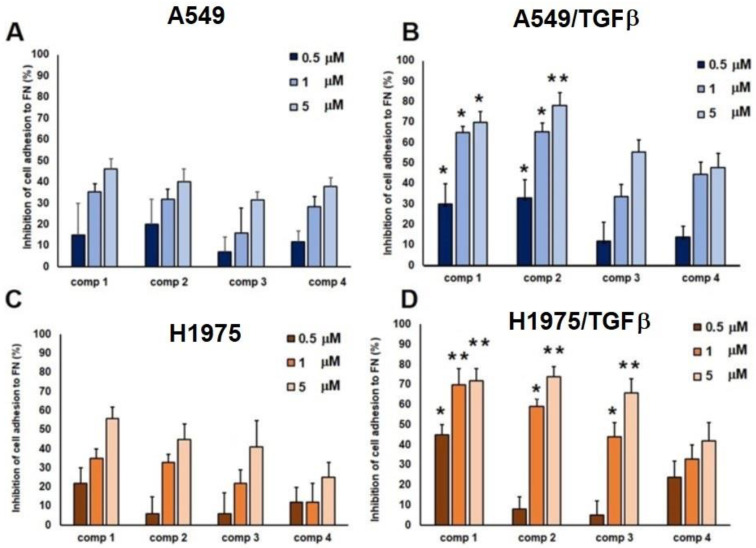
Inhibition of A549 (**A**,**B**) and H1975 (**C**,**D**) cell adhesion to FN (10 μg/mL) in the presence of compounds **1**–**3**, or *c*(AmpLRGDL) **4** (2 h). Cells were grown in standard conditions (**A**,**C**) or were previously exposed to TGF𝛽 (10 ng/mL) for 48 h (**B**,**D**). The inhibitory activity was calculated as the percentage of cell adhesion to FN in untreated cells and was expressed as mean ± SEM. Experiments were carried out in triplicate. * *p* < 0.05 and ** *p* < 0.0002 vs equimolar concentrations of compound **4**, by one-way ANOVA followed by Tukey’s multiple comparison test.

**Figure 4 ijms-24-01475-f004:**
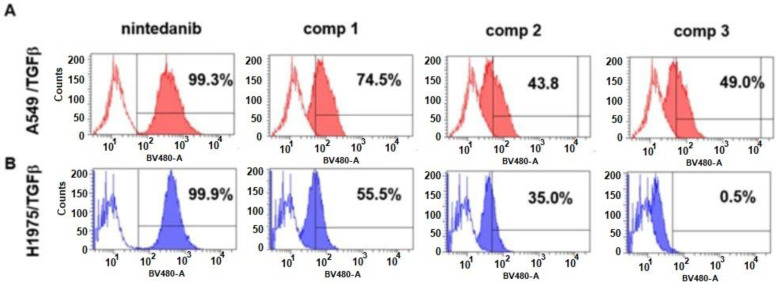
(**A**) Representative images of internalization of conjugates **1**–**3** or nintedanib as assessed by flow cytometry measurements. Fluorescence intensity (FacScan FLT1/BV405ex480em-A) in TGF𝛽-treated A549 cells (**A**) and TGF𝛽-treated H1975 cells (**B**) exposed for 24 h to nintedanib or conjugates **1**–**3** at 5 µM concentrations. Percentages indicate fluorescence (BV480-A)-positive cells from three independent experiments.

**Figure 5 ijms-24-01475-f005:**
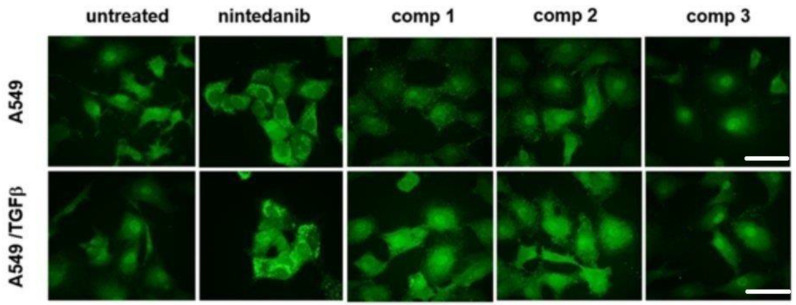
Representative images of fluorescence confocal analysis of A549 and TGF𝛽-treated A549 cells exposed to compounds **1**–**3** or nintedanib (5 μM) for 24 h. After incubation, cells were fixed and processed for fluorescence confocal imaging (Emission at 580 nm). Image dimensions: 180 × 180 μm, scale bar 50 μm. Experiments were performed at least three times.

**Figure 6 ijms-24-01475-f006:**
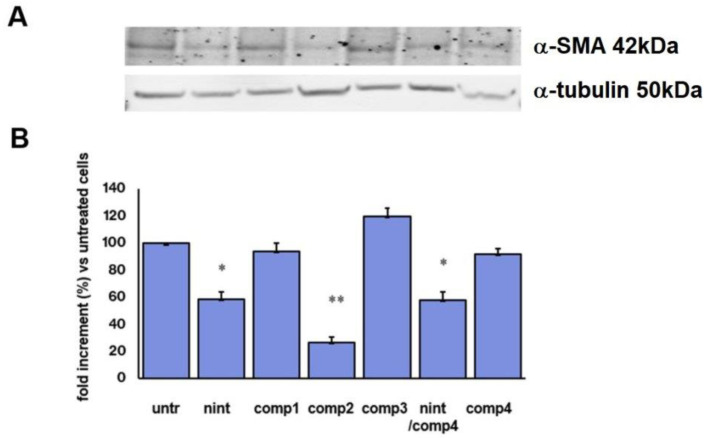
Expression of α-smooth muscle actin protein in TGF*β*-treated A549 cells grown in the presence of nintedanib (nint), compounds **1**–**3**, the combination of compound **4** with nintedanib, or compound **4** for 24 h (5 µM). (**A**) Representative Western blot analysis, (**B**) densitometric analysis of α-smooth muscle actin protein expression. Data are representative of three independent experiments (*n* = 4). * *p* < 0.05, and ** *p* < 0.001 vs untreated. Data are expressed as means ± SEM of fold increment compared to untreated cells normalized to α-tubulin.

**Figure 7 ijms-24-01475-f007:**
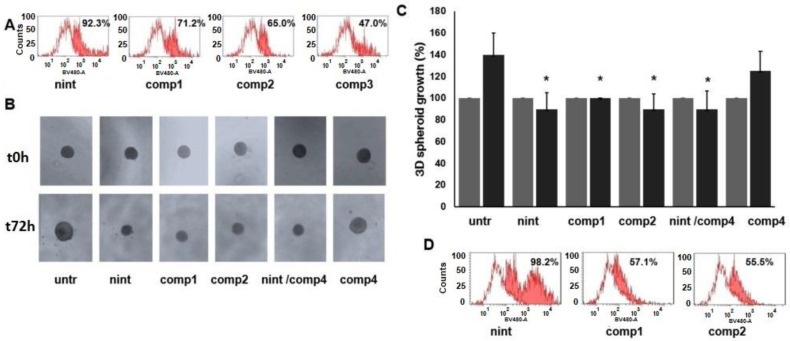
Inhibition of tumor spheroid growth. Hanging drop tumor spheroids were prepared starting from 1:1 mixed cell population (1000 cells/25μL) of TGF𝛽-treated A549 cells and murine L929 fibroblasts. (**A**) 3D spheroid internalization after the exposure to nintedanib, compounds **1**, **2**, or **3** (at 5 μM). (**B**) Representative images of inhibition of 3D spheroid growth after the exposure to nintedanib, compounds **1**, **2**, the combination of compound **4** with nintedanib or compound **4**. (**C**) Quantitative analysis of inhibition of 3D growth and (**D**) 3D spheroid internalization of the compounds after 72 h exposure. Data are representative of three independent experiments (*n* = 6) and expressed as means ± SEM. * *p* < 0.05 vs. untreated spheroids.

**Figure 8 ijms-24-01475-f008:**
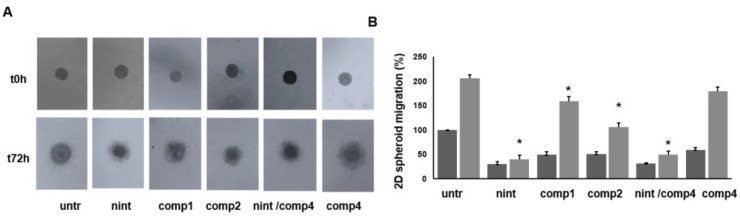
Inhibition of tumor spheroid flat surface migration. Hanging drop tumor spheroids were allowed to spread on a 2D fibronectin-coated flat surface. (**A**) Representative images of flat surface 2D migration of tumor spheroids after the exposure to nintedanib, compounds **1, 2**, the combination of compound **4** with nintedanib, or compound **4**. (**B**) Quantitative analysis of 2D migration. Data are representative of three independent experiments (*n* = 6) and expressed as means ± SEM. * *p* < 0.05 vs. untreated spheroids.

**Figure 9 ijms-24-01475-f009:**
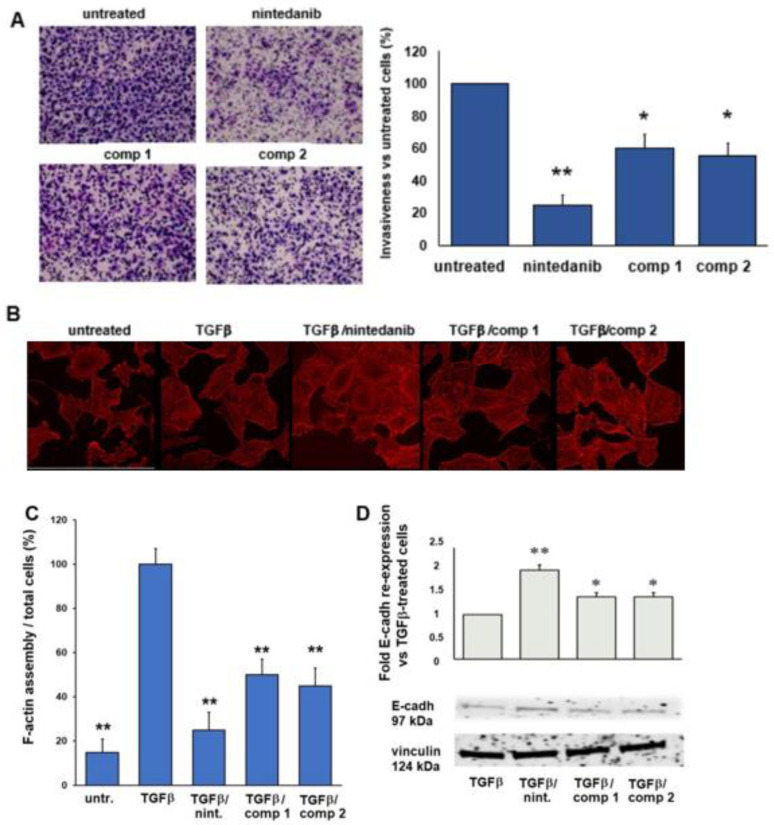
(**A**) Representative images of TGF𝛽-treated A549 cells invasiveness after growth in the presence of compounds **1**, **2**, or nintedanib at 5 μM for 24 h (left panel). Cells were allowed to migrate towards fibronectin-coated filters (10 μg/μL). Right panel, quantification of invasiveness. Data are percentages of migrated cells compared to untreated cells and expressed as mean value ± SEM of at least three independent experiments. * *p* < 0.05 and ** *p* < 0.001 vs. untreated cells. (**B**) Representative images of fluorescence confocal analysis of TGF𝛽-treated A549 cells exposed to nintedanib or compounds **1** and **2** at 5 μM for 24 h. After incubation, cells were fixed and processed for fluorescence confocal imaging. The cell cytoskeleton was stained with rhodamine-conjugated phalloidin. Image dimensions: 180 × 180 μm, scale bar 50 μm. Experiments were performed at least three times. (**C**) Quantification of disassembled cytoskeleton expressed as the percentage of cells with point-like fluorescence signals/total cells per field. Data are representative of three independent experiments. * *p* < 0.05, and ** *p* < 0.001 vs. TGF*β* treated cells. (**D**) E-cadherin re-expression, in TGF𝛽-treated A549 cells grown in the presence of nintedanib or compounds **1** and **2** at 5 μM for 24 h. Left panel, representative Western blot of E-cadherin protein expression. Right panel, densitometric analysis of E-cadherin re-expression in percentages, * *p* < 0.05, and ** *p* < 0.001 vs untreated cells.

**Figure 10 ijms-24-01475-f010:**
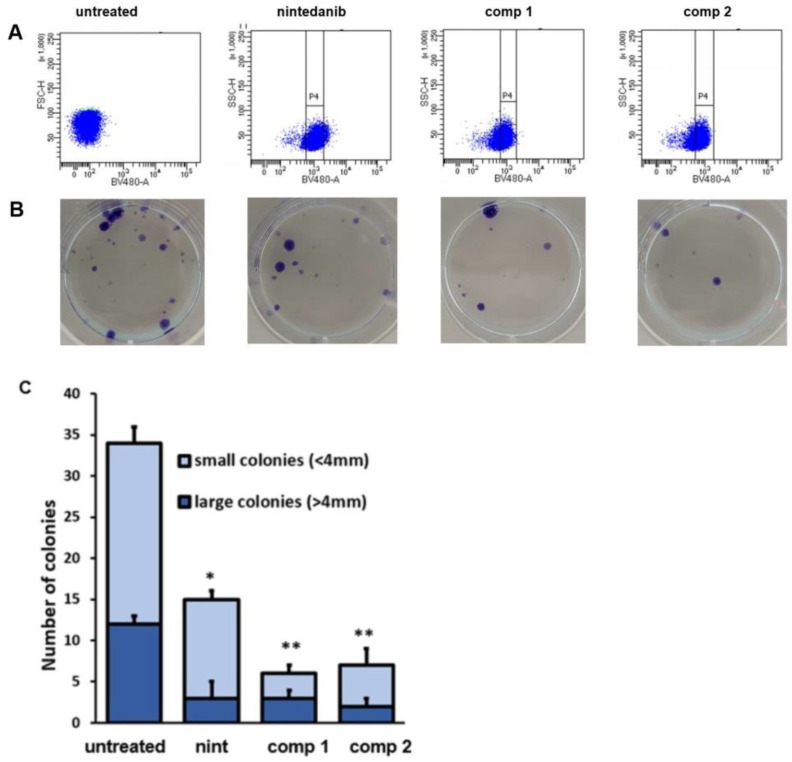
Inhibition of clonogenic activity. TGF𝛽-treated A549 cells were grown in the presence of nintedanib or compounds **1** or **2** for 24 h. (**A**) After incubation, cells were sorted (BDMelody) and isolated cells from the selected population (indicated as P4) were seeded at low density (1000 cells/6 well petri dishes). (**B**) Number of colonies obtained after 2 weeks growth in standard media. (**C**) Quantification of colonies. Data are representative of three independent experiments. * *p* < 0.05 and ** *p* < 0.001 vs. untreated cells.

## Data Availability

Data is contained within the article or Appendix A.

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
