# Peer review of "Nintedanib-αVβ6 Integrin Ligand Conjugates Reduce TGFβ-Induced EMT in Human Non-Small Cell Lung Cancer"

_ijms, 2023, doi:10.3390/ijms24021475_

Round 1

Reviewer 1 Report

This manuscript is very interesting but it presents some flaws that must be resolved. In particular:  

Lines 36-38: It deserves to be specified that TGFB can induce EMT downregulating tight junctions and increasing cell proliferation and survival (see PMID: 24768095, 26708185 ). Moreover, TGFB is an important ECM stimulator inducing collagen expression, a key component of ECM (see PMID: 32006713). These are three important processes to be highlighted since they play a crucial role in EMT. 

Figure 5: Scale Bar or Magnitude must be reported

Figure 6: Molecular weights must be shown

Figure 9: this figure is unreadable. Improve image quality

Figure 10A: improve image quality

4.3. Determination of αvβ6 expression: antibodies dilution must be reported

4.10. Invasion assay: which chemoattractant for migration did the authors use in this assay? was FCS present in both chambers or just in the lower chamger? this must be stated in this section

An accurate revision of syntax and punctuation is recommended

Author Response

Response to the 1st Reviewer

Thank you for your efforts, comments, and suggestion.

  • Lines 36-38: It deserves to be specified that TGFB can induce EMT downregulating tight junctions and increasing cell proliferation and survival (see PMID: 24768095, 26708185). Moreover, TGFB is an important ECM stimulator inducing collagen expression, a key component of ECM (see PMID: 32006713). These are three important processes to be highlighted since they play a crucial role in EMT.

Lines 42-49. We have gladly accepted reviewer’s suggestion and have enriched the bibliography as indicated.

  • Figure 5: Scale bar and magnitude have been reported in Figure legend
  • Figure 6. We added the molecular weights in the figure.
  • Figure 9. We provided to enlarge the different parts of the Figure.
  • Figure 10.  We provided to enlarge the different parts of the Figure.
  • Section 4.3. We reported the antibody dilution used in the cytofluorimetric assay.
  • Section 4.10. We clarified the experimental procedures used in the invasion assay. The assay was conducted in the absence of any chemoattractant.

Finally, we took care of providing an accurate revision of the English language and punctuation, also with the help of a mother-tongue English collaborator.

Best regards,

Francesca Bianchini, PhD

Professor of Pathology,

Department of Clinical and Experimental Biomedical Science

University of Florence, Italy

Reviewer 2 Report

In this study the authors reported 3 new molecular conjugated, compounds 1-3, where a selective αvβ6 receptor ligand was linked to nintedanib. These synthetized compounds were able to inhibit NSCLC cells adhesion to fibronectin and were internalized into αvβ6-expressing cells. Specifically, they found that two of the synthesized compounds (1 and 2) were able to block 2D and 3D tumor cell growth and reduce the clonogenic ability of TGFβ-treated lung cancer cell lines and other aspects involved in EMT phenotype. The goal of this paper is interesting however, some concerns are raised. Please see the comments below:

1.            Introduction: the authors should better explain the role of Arg-Gly-Asp sequence.

2.            In order to improve the reading flow, the authors are suggested to report the rationale that has driven each experiment.

3.            Figure 1: the authors should describe in the result section the main chemical features of the conjugated compounds 1-3. Moreover, they did not refer to the unconjugated ligand 4: how was it obtained? What is/are the vehicle/s to dissolve the synthetized compounds?

4.            Figure 2: The description of the results was not clear. The authors are suggested to better explain the data they have obtained and to adequatly report in the main text which graph in the Figure refers to A549 and H1975 cells. Moreover, the authors should explain the rationale for choosing two NSCLC cell lines

5.            Figure S2: What was the rationale underlying the evaluation of the expression of ?v?3 in A549 and H1975 human NSCLC cells? Moreover, why has ?v?3 expression in K562, SSM2c and L929 cells not been evaluated?

6.            The authors wrote: “In order to extend our experimental protocol, we evaluated the αvβ6 integrin expression on 111 human melanoma cells and murine fibroblasts”. The authors should indicate what ‘to extend our experimental protocol’ means. Moreover, why did they use murine fibroblasts? I believe that, according to the other human cell lines, it would have been better to use human fibroblasts, e.g., human primary fibroblast (ATCC® PCS-201-012™).

7.            Figure 3: The authors need to show the results obtained from untreated cells and cells exposed to the sole TGFβ. It has to be noted that, although they stated: “As shown in Figure 3, all compounds inhibited cell adhesion in a dose-related manner”, no statistical differences were reported in Figure 3A-3C (or they were missed?). In my opinion, the obtained results should be addressed to better detail the following issues:  1. The authors did not adequately report in the main text which graph in the Figure refers to A549 and H1975 cells (in the presence or not of TGFβ); 2. The compound 3 was able to inhibit cell adhesion in H1975cells. 3. The differences between A549 and H1975 in terms of response to the conjugated compounds were not stressed out.

8.            Figure S2E: As reported above, the results obtained from untreated cells and cells exposed to the sole TGF? are missing. The authors are also suggested to report in the main text which graph in the Figure refers to of A549 and TGF?-treated A549.

9.            Figure 4: The description of the obtained results is completely missing. The superficiality in the description of the results makes this research paper difficult to follow and understand.

10.          Figure 5: To confirm the data reported in Figure 4B, S4B and S5B, confocal microscopy analysis should be performed on H1975, SSM2c and L929 cells.

11.          Figure S6: According to the previous reported results, to explore whether the internalization of the compounds was affected by the exposed concentration, the authors should have chosen 0.5-1-5 µM concentration range. The authors are suggested to explain why they choose to test 10 µM.

12.          Figure S8: The authors need to indicate the tested concentrations for Nintedanib, compound 1-2 and 3. The results obtained from MTT assay of TGF?-treated H1975 cells (right panel) were missing.

13.          Figure 6: The authors need to show the expression of aSMA in A549 cells not treated with TGF?. Did ‘Untreated cells’ mean not treated A549 (sole cell medium) or TGF?-treated A549?

14.          Figure 7: The authors also found that spheroid growth was inhibited significantly after the treatment with compounds 1 and 2. Why didn’t the authors evaluate the impact of compounds 3 on 3D growth? Moreover, did the authors have data from spheroids obtained with H1975 cells?

15.          Line 329-335: This part should be included in the result section, where, as stated above, the description of the synthetized compounds was quite superficial

16.          Line 358-361: According to the comment 7, the authors should underlie that the compound 3 was efficient at inhibiting cell adhesion in H1975 cells. What could be the reason of this difference between A549 and H1975?

17.          Line 368, the authors stated “Cytofluorimetric assay confirmed that compounds 1 and 2 are internalized more efficiently than compound 3”. This is true for H1975 cells, but according to the results reported in Figure 4, the percentual internalization of compound 3 (49%) was similar to what observed for compound 2 (43.8%) in TGF?-treated A549.

18.          A section for the limitations/strength of this study should be reported

19.          Material&Methods: Cell culture conditions for H1975 and SSM2c were missing. The authors should explain how western blotting analyses were performed.

20.          The authors should give more emphasis to the impact of the obtained results in lung cancer context

21.          Typing errors should be fixed (e.g.: line 262; line 585)

Minor comments:

-Figure S3-S4-S5: the authors are suggested to include in the graphs the percentual expression of ?v?6 and the percentual internalization of conjugates 1-3 or nintedanib.

-Because the authors performed in vitro experiments, it could be better to use ‘concentration’ instead of ‘dose’.

-Figure S2E: The concentration unit (μM) was missed in the Figure

-Line 370: The authors should refer also to Figure 5.

Author Response

Response to the 2nd Reviewer

Thank you for your efforts, comments, and suggestions

  1. Introduction: the authors should better explain the role of Arg-Gly-Asp sequence.

We improved the introduction and clarified the role of RGD-recognizing integrin in cancer progression. 

  1. In order to improve the reading flow, the authors are suggested to report the rationale that has driven each experiment.

We provided a brief rationale of each experiment.

  1. Figure 1: the authors should describe in the result section the main chemical features of the conjugated compounds 1-3. Moreover, they did not refer to the unconjugated ligand 4: how was it obtained? What is/are the vehicle/s to dissolve the synthetized compounds?

 The main chemical features of the conjugates 1-3 and the reference to the preparation of unconjugated ligand 4 were added in the result section in paragraph 2.1.

The synthetized compounds 1-3, unconjugated ligand 4, and nintedanib were dissolved in DMSO at 10 mM concentration, as reported in section 4.1.

  1. Figure 2: The description of the results was not clear. The authors are suggested to better explain the data they have obtained and to adequately report in the main text which graph in the Figure refers to A549 and H1975 cells. Moreover, the authors should explain the rationale for choosing two NSCLC cell lines

We provided an accurate description of the results by adding referring letters to each panel. In the discussion, we provided the explanation for choosing the two NSCLC cell lines used.

 5. Figure S2: What was the rationale underlying the evaluation of the expression of αvβ3 in A549 and H1975 human NSCLC cells? Moreover, why has αvβ3 expression in K562, SSM2c and L929 cells not been evaluated?

In the discussion, we added an extensive explanation of the rationale underlying the evaluation of the αvβ3 in cancer cells. Moreover, we added specific references indicating the evaluation of αvβ3 expression in K562, L929 and SSM2c cells in previously published papers (section 2.2).

  1. The authors wrote: “In order to extend our experimental protocol, we evaluated the αvβ6 integrin expression on 111 human melanoma cells and murine fibroblasts”. The authors should indicate what ‘to extend our experimental protocol’ means. Moreover, why did they use murine fibroblasts? I believe that, according to the other human cell lines, it would have been better to use human fibroblasts, e.g., human primary fibroblast (ATCC® PCS-201-012™).

We understand that co-cultures using murine and human cells are rare and uncommon, however, the main scope of the procedure was to recreate a complex multicellular environment in which integrins were playing their critical role. Since integrins of different species share strong similarities and functions, we chose to create co-cultured spheroids using L929 murine fibroblasts as a proof of concept. 

  1. Figure 3: The authors need to show the results obtained from untreated cells and cells exposed to the sole TGFβ. It has to be noted that, although they stated: “As shown in Figure 3, all compounds inhibited cell adhesion in a dose-related manner”, no statistical differences were reported in Figure 3A-3C (or they were missed?). In my opinion, the obtained results should be addressed to better detail the following issues: 1. The authors did not adequately report in the main text which graph in the Figure refers to A549 and H1975 cells (in the presence or not of TGFβ); 2. The compound 3 was able to inhibit cell adhesion in H1975cells. 3. The differences between A549 and H1975 in terms of response to the conjugated compounds were not stressed out.

1- We provided the correct description of the Figure 3 by adding referring titles to each panel.

2- We explained more in details the results and the statistical significance which is related to unconjugated ligand (section 2.3).

3- In the discussion, we explained the differences in the inhibition of adhesion in the two cell lines.

  1. Figure S2E: As reported above, the results obtained from untreated cells and cells exposed to the sole TGFβ are missing. The authors are also suggested to report in the main text which graph in the Figure refers to of A549 and TGF?-treated A549.

We provided to correct the Figure S2E, and the main text, accordingly.

  1. Figure 4: The description of the obtained results is completely missing. The superficiality in the description of the results makes this research paper difficult to follow and understand.

We took care to extensively describe the results (section 2.3, and discussion).

  1. Figure 5: To confirm the data reported in Figure 4B, S4B and S5B, confocal microscopy analysis should be performed on H1975, SSM2c and L929 cells.

The results obtained using the confocal analysis are very interesting. Even if the confocal analysis is an elegant way to show the internalization of the conjugates, it furnishes only qualitative information on a few cells. This procedure has been used to confirm what we observed with cytofluorimetric analysis. In fact, we preferred to investigate cellular internalization using the cytofluorimetric analysis, in different types of cells. This assay allowed us to quantify the internalization of the compounds in a large number of cells.

 11. Figure S6: According to the previous reported results, to explore whether the internalization of the compounds was affected by the exposed concentration, the authors should have chosen 0.5-1-5 µM concentration range. The authors are suggested to explain why they choose to test 10 µM.

The use of different range of concentration could be critical in obtaining significant results for the evaluation of different types of biological characteristics. Thus, we chose the 0.1-5 µM range for the evaluation of inhibition of adhesion. This low-concentration range, indeed, is quite similar to that used in the evaluation of the ligand affinity on isolated receptor. Conversely, cellular internalization required a higher concentration range. Indeed, we found that the internalization was markedly reduced at 1 µM already, especially of compound 3, and for this reason, we extended the concentration range to higher levels instead of lower levels. This assay was also useful to choose the concentration of the following biological assays since nintedanib moiety has to be internalized to inhibit the tyrosine kinase receptor.

 12. Figure S8: The authors need to indicate the tested concentrations for Nintedanib, compound 1-2 and 3. The results obtained from MTT assay of TGFβ-treated H1975 cells (right panel) were missing.

We provided the tested concentrations for the three compounds used in the MTT assay 

  1. Figure 6: The authors need to show the expression of aSMA in A549 cells not treated with TGFβ. Did ‘Untreated cells’ mean not treated A549 (sole cell medium) or TGFβ-treated A549?

Untreated cells mean: cells exposed to the sole TGFβ. We found a weak expression of α-SMA in naïve cells (in preliminary investigations, not shown), as also shown in the immunofluorescence assay (Figure 9B). 

  1. Figure 7: The authors also found that spheroid growth was inhibited significantly after the treatment with compounds 1 and 2. Why didn’t the authors evaluate the impact of compounds 3 on 3D growth? Moreover, did the authors have data from spheroids obtained with H1975 cells?

Compound 3 was not used in the inhibition of spheroid growth due to its global less efficacy in intracellular localization in different types of cells, and α-SMA inhibition in A549/ TGF?-treated. Thus, we preferred to perform the next biological assays only on the most promising compounds.

We did not evaluate the inhibition of spheroids from H1975 cells. 

  1. Line 329-335: This part should be included in the result section, where, as stated above, the description of the synthetized compounds was quite superficial

We added a more detailed description of the compounds in the result section in paragraph 2.1, as previously indicated in the response to suggestion #3 

  1. Line 358-361: According to the comment 7, the authors should underlie that the compound 3 was efficient at inhibiting cell adhesion in H1975 cells. What could be the reason of this difference between A549 and H1975?

As suggested, we added more details in the discussion. 

  1. Line 368, the authors stated “Cytofluorimetric assay confirmed that compounds 1 and 2 are internalized more efficiently than compound 3”. This is true for H1975 cells, but according to the results reported in Figure 4, the percentual internalization of compound 3 (49%) was similar to what observed for compound 2 (43.8%) in TGFβ-treated A549.

According to the reviewer’s suggestion, we changed the text and added a correct comment.

 18. A section for the limitations/strength of this study should be reported

At the end of the discussion, we went through the potential limitation of this study. 

  1. Material&Methods: Cell culture conditions for H1975 and SSM2c were missing. The authors should explain how western blotting analyses were performed.

The missing part has been integrated and more details have been added regarding Western blotting analyses. 

  1. The authors should give more emphasis to the impact of the obtained results in lung cancer context

We added a paragraph at the end of the discussion, in which we explained the potential role of dual targeting in lung cancer treatment. 

  1. Typing errors should be fixed (e.g.: line 262; line 585)

We took care of providing an accurate revision of the English language and punctuation.

Minor comments:

-Figure S3-S4-S5: the authors are suggested to include in the graphs the percentual expression of αvβ6 and the percentual internalization of conjugates 1-3 or nintedanib.

We added the percentage expression of αvβ6 and the percentage internalization of conjugates 1-3 or nintedanib.

-Because the authors performed in vitro experiments, it could be better to use ‘concentration’ instead of ‘dose’.

We substituted the term “dose” with the corrected term “concentration”.

-Figure S2E: The concentration unit (μM) was missed in the Figure

We added the concentration unit in the Figure.

-Line 370: The authors should refer also to Figure 5.

We added Figure 5 as reference.

Thank you for your kind assistance,

Best regards,

Francesca Bianchini, PhD

Professor of Pathology, Department of Clinical and Experimental Biomedical Science

University of Florence

Italy

Round 2

Reviewer 1 Report

the manuscript has been significantly improved and can be accepted in the present form 

Reviewer 2 Report

The manuscript could be published